# Visible Light Active Magnesium Silicate–Graphitic Carbon Nitride Nanocomposites for Methylene Blue Degradation and Pb$^{2+}$ Adsorption

Muhmmed Ali Alnassar, Abdulmohsen Alshehri and Katabathini Narasimharao *

Chemistry Department, Faculty of Science, King Abdulaziz University, P.O. Box 80203,
Jeddah 21589, Saudi Arabia
* Correspondence: nkatabathini@kau.edu.sa; Tel.: +966-538638994

**Abstract:** Magnesium silicate nanosheets (MgSiNS) and graphitic carbon nitride (g-C$_3$N$_4$) nanocomposites were produced by varying different weight percentages of g-C$_3$N$_4$. The obtained nanocomposites were characterized by various techniques such as X-Ray diffraction (XRD), Fourier transformed infrared spectroscopy (FT-IR), diffuse reflectance UV–vis spectroscopy (DR UV–vis), N$_2$-physisorption, transmission electron microscopy (TEM), and X-ray photon spectroscopy (XPS). The photocatalytic activities of the nanocomposites were measured using visible light irradiation to degrade methylene blue (MB) and Pb$^{2+}$ adsorption in aqueous solution. The ideal physicochemical properties such as porosity, band gap energy, and functional groups in the MgSiNS-GN20 composite (80% MgSiNS and 20 wt % of g-C$_3$N$_4$) offered high Pb$^{2+}$ adsorption (0.005 mol/g) and excellent MB degradation efficiency (approximately 93%) at pH 7 within 200 min compared to other composites. In addition, the influences of different reaction parameters such as the effect of pH, the load catalyst, and the concentration of MB and Pb$^{+2}$ ions were examined. The obtained results indicate that inexpensive and eco-friendly MgSiNS and g-C$_3$N$_4$ composites could be recycled several times, hence representing a promising material to purify water from both organic and inorganic contaminants.

**Keywords:** magnesium silicate; g-C$_3$N$_4$; nanosheets; photodegradation; methylene blue; visible light; Pb$^{2+}$ adsorption

## 1. Introduction

Water pollution from toxic and harmful pollutants, including organic toxins and inorganic heavy metal species, is a genuine challenge as it could endanger the environment [1]. Methylene blue (MB) is an organic dye that contains aromatic amine functional groups, which can be absorbed by the human body through ingestion, leading to dangerous levels of carcinogenicity and mutagenicity. The hindrance of sunlight access to aquatic life because of MB accumulation is considered another negative impact, as the recycling of oxygen in water will be ineffective [2]. Lead (Pb) is one of the toxic heavy metals responsible for soil, water, and atmospheric pollution. Pb could affect aquatic and human life adversely even at a low concentration [3]. Therefore, researchers are devoted to developing scientific methods such as precipitation, coagulation, ion exchange, biological treatment, and advanced oxidation processes for water purification from organic and inorganic contaminants such as MB and Pb metal ions [2–4]. Among the developed methods, adsorption and photocatalytic degradation have been commonly used in water treatment [5,6].

Nanoscale metal silicate materials attracted the interest of many researchers in recent years since these materials have versatile physicochemical properties [7,8]. Metal silicates are widely used as adsorbents and catalysts because they are inexpensive, environmentally friendly, and thermally stable at high temperatures [9,10], which are considered as essential aspects in industrial utilization. Liang et al. synthesized 2D layered zinc silicate materials and used them for the degradation of chlorophenol in aqueous solution. However, it was

observed that the thickness of layered metal silicates is quite high, leading to their low adsorption capacity [11]. In another report, Wu et al. synthesized calcium silicate hydrate nanosheets and utilized them as an adsorbent for heavy metal species such as $Cd^{+2}$, $Pb^{+2}$, etc. The synthesized calcium silicates were found to be effective for adsorption processes due to their high surface area (505 $m^2$ $g^{-1}$) [12].

Magnesium silicate powders were previously synthesized and used in ceramic, detergents, food, cosmetic, and pharmaceutical industries for different purposes [13]. It was reported that magnesium silicates play a crucial role in absorbing humidity and acting as an anticaking agent [13,14]. Nanoscale magnesium silicate and its composite were among various metal silicates that attracted interest in water treatment applications [15]. Haung et al. succeeded in synthesizing magnesium silicate hydrate nanoparticles in different ratios of Si/Mg and studied the role of surface charge in MB adsorption capacity. It was observed that adsorbents with a Si/Mg ratio of 1.75 at pH 10 effectively adsorbed the MB within 60 min [16]. Yuan et al. prepared magnesium silicate nanotubes using the hydrothermal method and observed that ethylene glycol played a critical role in obtaining nanotube morphology. These authors evaluated the ability of the magnesium silicate remove $Pb^{+2}$ and $Cd^{+2}$ ions [17]. In another report, Chen et al. applied magnesium silicates to eliminate radioactive U VI) ions and observed a removal percentage of roughly 73% in 120 min [18].

In recent years, g-$C_3N_4$ nanosheets gained much attention due to their importance in the development of supported catalysts and nanocomposite materials for the removal of organic and inorganic pollutants [19]. It was observed that g-$C_3N_4$ is an effective semiconductor for photocatalytic degradation of organic pollutants due to its unique thermal stability and optical properties [20]. Xu et al. synthesized g-$C_3N_4$ by mixing dicyandiamide and thiourea in different ratios, followed by calcination in a muffle furnace. The prepared g-$C_3N_4$ was successfully utilized to degrade MB and phenol molecules under visible light [21]. Liu et al. discovered a simple route to obtain yellowish g-$C_3N_4$ through thermal decomposition of urea at different temperatures starting from 400 to 550 °C for 3 h, and the obtained g-$C_3N_4$ showed excellent recyclability and photodegradation for MB under visible light [22]. The aim of this research was to combine the properties of two different nanomaterials featuring different functional properties, which can also be easily regenerated and reused for water purification. It is important that the synthesized nanocomposites do not contribute to secondary pollution. To achieve this goal, for the first time, we synthesized magnesium silicate and g-$C_3N_4$ nanocomposites to utilize them for the removal of inorganic ($Pb^{2+}$ adsorption) and organic pollutants (MB degradation) under visible light. We also aimed to correlate the physicochemical properties of synthesized magnesium silicate–g-$C_3N_4$ nanocomposites with their ability to remove both organic and inorganic pollutants under ambient conditions.

## 2. Results and Discussion

The phase identification and crystal size of the prepared samples were analyzed by XRD. The XRD patterns of all synthesized samples, including pure magnesium silicate nanosheet (MgSiNS), graphitic carbon nitride (g-$C_3N_4$), and the MgSiNS-GN nanocomposites, are shown in Figure 1A. The synthesized bulk MgSiNS sample exhibited broad reflections at 19.81°, 28.05°, 35.81°, and 60.91°, which can be attributed to (020), (1-12), (-133), and (-332) planes of monoclinic $Mg_3Si_4O_9(OH)_{10}$ phase [23,24]. The presence of low-intensity and broad XRD reflections of MgSiNS is an obvious indication that the sample possessed small-sized crystallites. On the other hand, the bulk g-$C_3N_4$ sample exhibited two reflections at 12.9° and 27.3°, which could be indexed to (100) and (002) planes of g-$C_3N_4$ phase. The XRD pattern of the prepared bulk g-$C_3N_4$ is in agreement with JCPDS file no. 87-1526 for the hexagonal g-$C_3N_4$ phase [25]. The MgSiNS-GN5 and MgSiNS-GN10 nanocomposites showed similar reflections to the bulk MgSiNS sample. It is interesting to note that reflections due to the g-$C_3N_4$ phase did not appear in these two nanocomposite samples, possibly due to a low g-$C_3N_4$ concentration and its incorporation in the magnesium silicate inner layers. On the other hand, MgSiNS-GN15 and MgSiNS-GN20

and MgSiNS-GN25 nanocomposite samples exhibited reflections due to both g-C$_3$N$_4$ and MgSiNS phases. The intensity of reflections due to the g-C$_3$N$_4$ phase was increased with the increase in its loading from 15 wt % to 25 wt %. Interestingly, a small shift in the position of reflection corresponding to the (200) plane of g-C$_3$N$_4$ (from 27.3° to 27.7°) was observed in the XRD patterns of MgSiNS-GN15, MgSiNS-GN20, and MgSiNS-GN25 samples. This observation indicates that the crystal lattice of g-C$_3$N$_4$ was compressed due to its presence in layers of magnesium silicate nanosheets. The crystallite sizes of the MgSiNS and g-C$_3$N$_4$ phases were calculated using the Scherrer equation [26] and the data are presented in Table 1. The results indicate that the crystallite sizes of the MgSiNS and g-C$_3$N$_4$ phases in composite samples are slightly low compared to parent materials, probably due to the ultrasonic treatment.

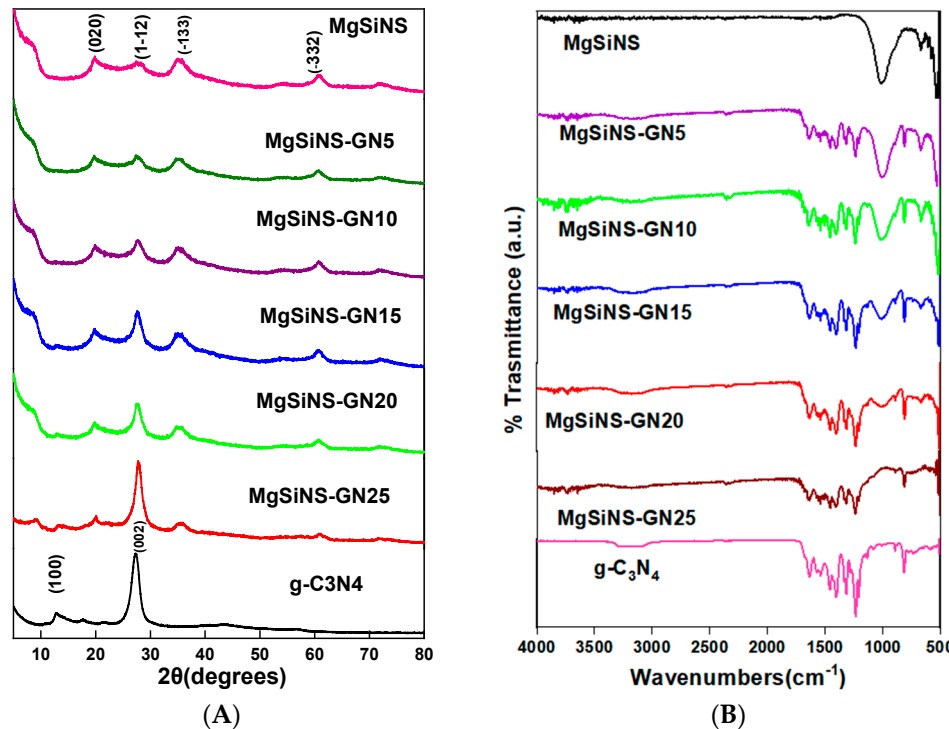

**Figure 1.** (**A**) XRD patterns and (**B**) FT−IR spectra of synthesized samples.

**Table 1.** BET surface area and the pore size, pore volume, and crystal sizes of MgSiNS and g−C$_3$N$_4$ phases.

| Sample | BET Surface Area (m²/g) | Average Pore Size (nm) | Pore Volume (cc/g) | Crystal Size (nm) | |
|---|---|---|---|---|---|
| | | | | **MgSiNS** | **g-C$_3$N$_4$** |
| MgSiNS | 294 | 3.4 | 0.103 | 2.50 | - |
| g−C3N4 | 121 | 3.8 | 0.132 | 4.16 | 5.53 |
| MgSiNS−GN5 | 186 | 4.8 | 0.077 | 4.19 | - |
| MgSiNS−GN10 | 159 | 4.3 | 0.060 | 4.41 | - |
| MgSiNS−GN15 | 116 | 3.9 | 0.040 | 5.34 | 5.12 |
| MgSiNS−GN20 | 108 | 3.8 | 0.037 | 5.41 | 5.03 |
| MgSiNS−GN25 | 88 | 3.9 | 0.009 | - | 4.89 |

To determine the functional groups and structural features of the synthesized materials, FT-IR spectra of the samples were obtained. The FT-IR spectra for all the samples are shown in Figure 1B in the range of 500–4000 cm⁻¹. The FT−IR spectrum of bulk MgSiNS sample showed IR absorption peaks at 560, 675, 684, and 1010 cm⁻¹, which could be assigned to Mg−O vibration, Si-O vibration, Mg-O-Si overlapping symmetric stretching vibration, and Si-O-Si asymmetric stretching vibration, respectively [24,27–29]. A less intense peak

observed at 3700 cm$^{-1}$ is due to the vibration of Mg-OH groups [30]. The FT−IR spectrum of the bulk g−C$_3$N$_4$ sample exhibited two peaks at 815 and 888 cm$^{-1}$ due to bending vibrations of triazine rings [31]. IR absorption peaks at 1229 and 1319 cm$^{-1}$, which could be attributed to secondary amine 2° (C−N) and tertiary amine 3° (C−N) vibrations in triazine rings, were also presented [25,32]. In addition, peaks at 1570 and 1630 cm$^{-1}$ due to stretching vibration modes of C = N groups and broad peaks in the range of 3029–3347 cm$^{-1}$ related to N-H vibration modes also appeared [33]. The MgSiNS-GN5, MgSiNS−GN10, MgSiNS−GN15, and MgSiNS−GN20 nanocomposites show IR peaks due to functional groups of both MgSiNS and g-C$_3$N$_4$ structures, indicating the formation of a composite between the two structures. Interestingly, the MgSiNS−GN25 sample showed a similar spectrum to the pure g-C$_3$N$_4$ sample, except for the peak at 3700 cm$^{-1}$ assigned to Mg−OH vibration; this is possibly due to the complete surface coverage of MiSiNS by g-C$_3$N$_4$.

To investigate the morphology and size of the synthesized samples, TEM analyses of bulk MgSiNS, g-C$_3$N$_4$, and representative composites (MgSiNS-GN5 and MgSiNS-GN15) were performed (Figure 2). The TEM image of bulk g-C$_3$N$_4$ exhibited many enfolded and wrinkled thin sheets; on the other hand, the pure MgSiNS sample showed the presence of relatively thick (15 nm) sheets, revealing the successful synthesis of MgSiNS and g-C$_3$N$_4$ with sheet morphology. The TEM images of MgSiNS-GN5 and MgSiNS-GN20 nanocomposites also show sheet morphology composed of both MgSiNS and g-C$_3$N$_4$ structures; however, the increase in g-C$_3$N$_4$ loading to 20 wt % resulted in the formation of large voids across the perimeter of MgSiNS nanosheets. The TEM results show evidence that the incorporation of g-C$_3$N$_4$ causes disorder in the MgSiNS structure.

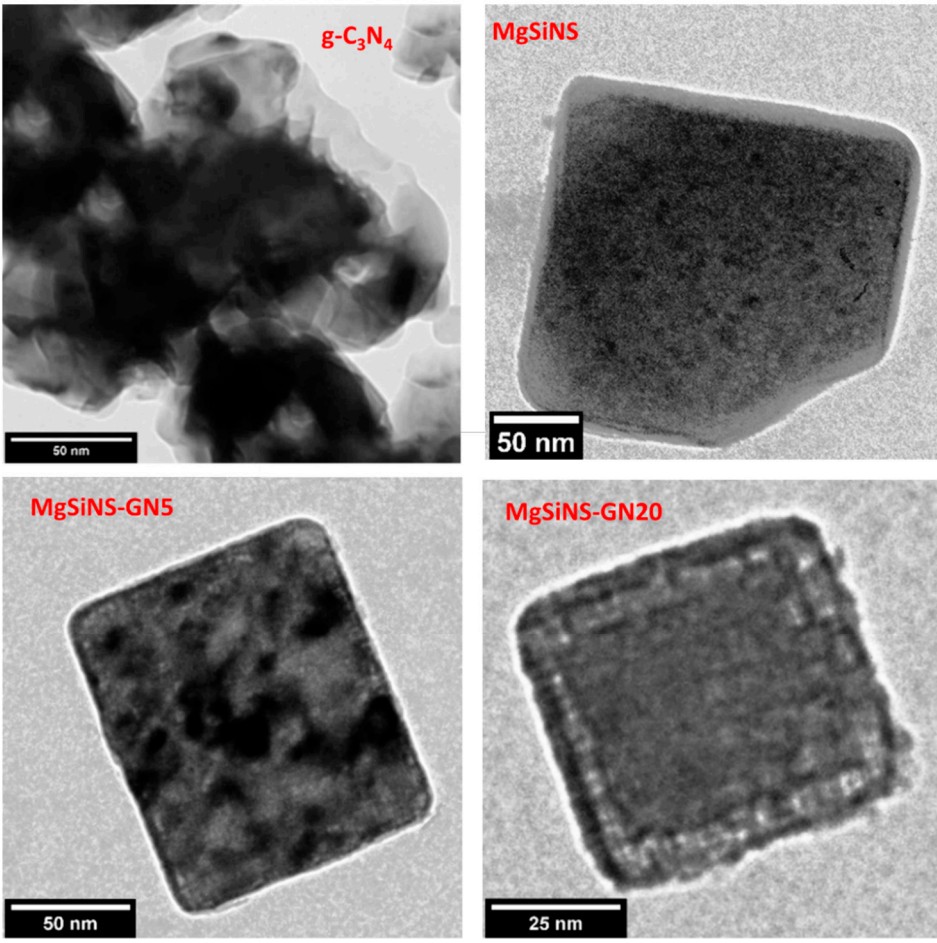

**Figure 2.** TEM images of MgSiNS, g-C$_3$N$_4$, MgSiNS-GN5, and MgSiNS-GN20 samples.

In order to study the light absorption properties and determine the band gap energy of the synthesized samples, DR UV–vis spectroscopy was used. The DR UV–vis spectra for all the samples are shown in Figure 3. The absorption band at 250 nm in the case of the bulk MgSiNS sample could be assigned to the metastable state developed between the valence band and the conduction band. The DR UV–vis spectrum of the MgSiNS sample did not show any absorption peak in the visible region. The pure g-C$_3$N$_4$ sample exhibited three absorption peaks at 255, 320, and 380 nm, due to the N=C functional groups of 1,3,5-triazine rings, uncondensed C=O, and the terminal N-C groups of g-C$_3$N$_4$, respectively [33]. The band tailing from 400 to 450 nm revealed that the synthesized g-C$_3$N$_4$ sample absorbed the light in the visible region. It is interesting to note that MgSiNS-GN nanocomposites exhibited light absorption properties similar to g-C$_3$N$_4$, including the sample containing 5 wt % of g-C$_3$N$_4$ loading, indicating that the electronic properties of MgSiNS dramatically changed due to its interaction with g-C$_3$N$_4$. However, the absorption edges of the composites are different from each other, and they were influenced by the g-C$_3$N$_4$ loading. The determination of band gap energy (E$_{bg}$) is important for any semiconductor material. The band gap energy of the samples was calculated according to the Kubelka–Munk formula by plotting the graph of the square of the Kubelka–Munk function $[\alpha(h\nu)]^2$ versus energy (Figure S1) [34]. The bulk g-C$_3$N$_4$ sample possessed band gap energy of 2.89 eV, while the bulk MgSiNS sample exhibited 4.36 eV. It is clear that the band gap energy of MgSiNS-GN nanocomposites is similar to the pure g-C$_3$N$_4$ sample, and interestingly, the band gap of nanocomposites slightly decreased with the increase in the loading of g-C$_3$N$_4$. The lowest band gap energy was observed in the case of the MgSiNS-GN20 sample (2.80 eV), probably due to optimum g-C$_3$N$_4$ loading.

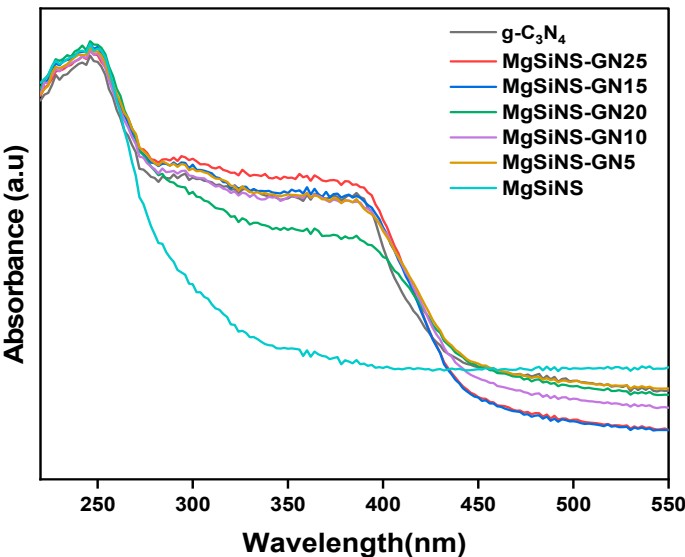

**Figure 3.** DR UV–vis absorption spectra for all the prepared materials.

To examine the textural properties of the synthesized samples, N$_2$-physisorption measurements were performed. The N$_2$ adsorption–desorption isotherms and pore size distribution patterns for all the samples are shown in Figure 4A,B, respectively. The N$_2$ adsorption–desorption isotherms for all samples appeared as type IV according to IUPAC classification [35], indicating the presence of mesopores in the samples. However, it is clear from Figure 4A that the samples exhibited different types of hysteresis loops; the isotherm of the bulk MgSiNS sample showed H2-type hysteresis, which is distinctive compared to other samples exhibiting the H4 type. The presence of H2-type hysteresis features a narrow pore distribution in the pore neck at low pressure, whereas H4-type hysteresis has multilayers and capillary condensation at high pressure [35]. The BET surface area for bulk MgSiNS is 294 m$^2$/g and its pore diameter and pore volume are 1.70 nm and 0.103 cc/g, respectively. On the other hand, the bulk g-C$_3$N$_4$ sample exhibited BET surface area of

121 m$^2$/g with pore size of 1.90 nm and pore volume of 0.132 cc/g. It is observed that the surface area of the composites is influenced by the g-C$_3$N$_4$ loading—the surface area, pore size, and pore volume decrease with the increase in g-C$_3$N$_4$ loading (Table 1).

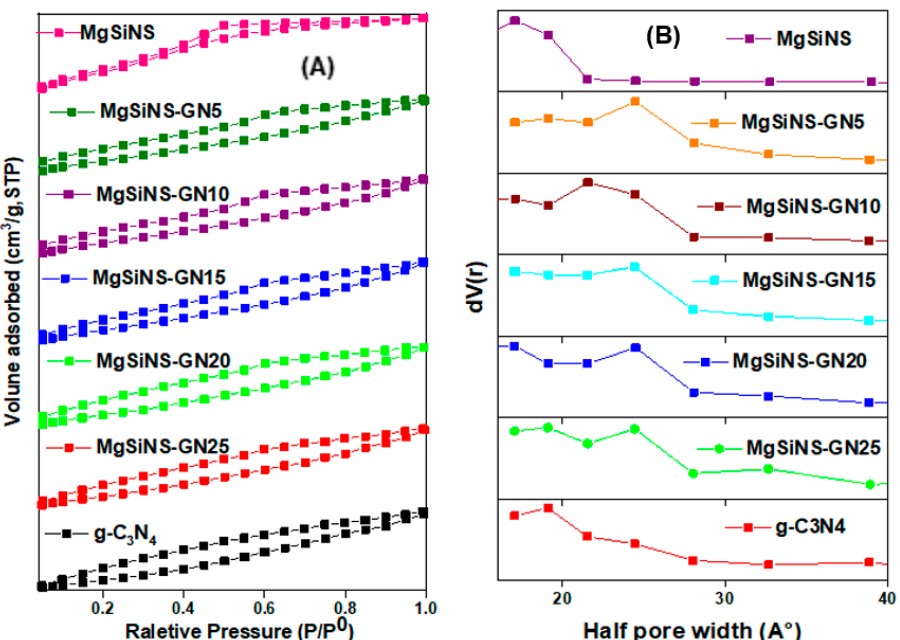

**Figure 4.** (**A**) N$_2$ adsorption–desorption isotherms and (**B**) pore size distribution patterns of the samples.

The surface chemical states and the compositions of the synthesized samples were measured using XPS spectroscopy. The XPS spectra for pure MgSiNS, g-C$_3$N$_4$, and two representative composites (MgSiNS-GN5, MgSiNS-GN20) were measured. The full XPS survey spectra of for the investigated samples are shown in Figure 5. All the samples show XPS peaks due to the presence of C, Mg, O, and Si elements. Interestingly, the existence of C*1s* in the pure MgSiNS sample is due to residual carbon from the organic template during the preparation. The deconvoluted XPS spectra for MgSiNS and g-C$_3$N$_4$ samples are shown in Figures S2 and S3. A single broad Mg*1s* peak at 1304.0 eV, which could be attributed to Mg-O [36], and a small Si*2p* peak around 102.7 eV, due to Si in SiO$_4$ structural units [37], were observed. However, the O*1s* component for the MgSiNS sample shows the presence of four peaks at 531.3, 531.8, 531.9, and 532.7 eV corresponding to Si-O, Si-O-Mg, Mg-OH, and Si-O-Si, respectively [38–40]. Three different C*1s* peaks at 288.5 eV, 284.5 eV, and 286.6 eV could be ascribed to N-C=N groups in *sp*$^2$ hybridized carbon atom of aromatic ring, C-C bond, and C-O groups in the g-C$_3$N$_4$ structure [22]. The deconvoluted N*1s* XPS spectrum of the pure g-C$_3$N$_4$ sample exhibited three peaks. The major peak at 398.5 eV is due to the N atom in C-N-C groups, and the other peaks at 400.1 eV and 401.0 eV could be ascribed to the N-©$_3$ and C-N-H group, respectively, in g-C$_3$N$_4$ [41]. Interestingly, a single O*1s* peak was observed at 532.3 eV, corresponding to C-O functional groups presented on the surface of g-C$_3$N$_4$ sheets. The observed XPS results are in accordance with the reported results in the literature. The XPS peaks for two selected nanocomposites (MgSiNS-GN5 and MgSiNS-GN20) were also investigated, and all the peaks for different chemical states (C*1s*, N*1s* and Si*2p*) were observed at the same binding energy position as their parent samples. However, it is noticed that the there is a minor shift in the binding energy position in Mg*1s* and O*1s* peaks, as shown in Figure 6; this is possibly due to the interaction between MgSiNS and g-C$_3$N$_4$ phases.

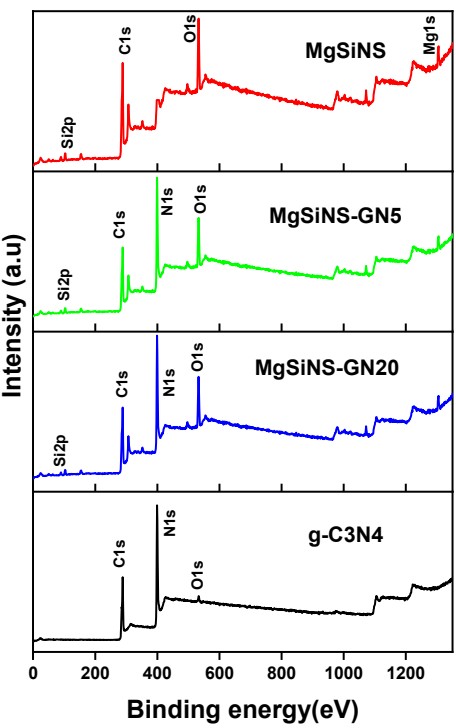

**Figure 5.** Full survey XPS spectra for bulk MgSiNS, g-C₃N₄, and two selected nanocomposites.

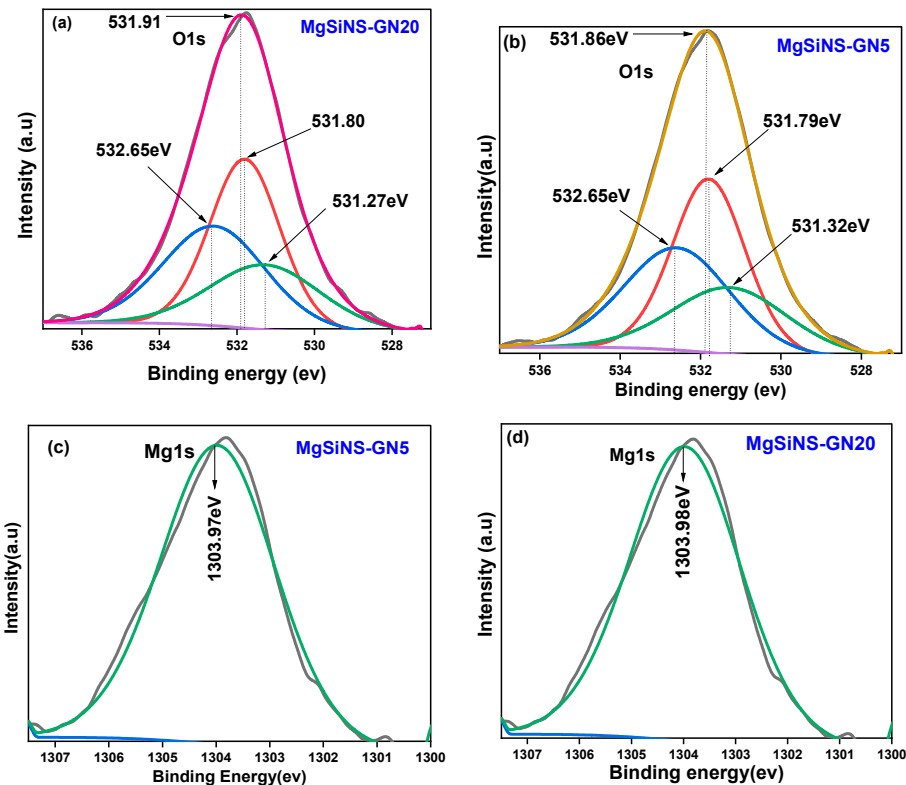

**Figure 6.** Deconvoluted XPS spectra for two selected nanocomposites O*1s* of (**a**) MgSiNS-GN20 and (**b**) MgSiNS-GN5, and Mg*1s* of (**c**) MgSiNS-GN5 and (**d**) MgSiNS-GN20.

All the prepared materials were tested as photocatalysts to degrade methylene blue (MB) dye under visible light irradiation. The photocatalytic degradation efficiency of materials over time was measured using visible light irradiation, shown in Figure 7. Initially,

the powder nanocomposite sample was added to MB solution, and the flask was kept in the dark for 30 min under constant stirring to examine the possible physical adsorption of MB on the surface of the photocatalyst. The catalysts did not show any significant (less than 4%) MB physical adsorption. The light source was switched on to start the photocatalytic experiments after quantification of the physically adsorbed MB on the catalyst surface. It was observed that the parent materials (MgSiNS and g-C$_3$N$_4$) offered low photocatalytic activity under visible light. When the reactant molecules and catalyst were exposed to visible light irradiation, a gradual increase in MB photodegradation efficiency was observed with time. The reaction was continued for 200 min, when the highest degradation efficiency was observed with all the synthesized samples. MgSiNS-GN20 followed MgSiNS-GN25, bulk g-C$_3$N$_4$, MgSiNS-GN15, MgSiNS-GN10, MgSiNS-GN5, and pure MgSiNS. The MgSiNS-GN20 and MgSiNS-GN25 composites acted almost similarly in degrading the MB within 200 min, reaching approximately 85%. However, the MgSiNS-GN25 sample exhibited the highest MB physical adsorption (57%) compared to other tested samples. The physical adsorption of MB takes the following order: MgSiNS-GN20 > MgSiNS-GN15 > MgSiNS-GN10 > MgSiNS-GN5 > MgSiNS > g-C$_3$N$_4$. Hence, the MgSiNS-GN20 composite sample was chosen to study the other photocatalytic reaction parameters, such as the effect of pH, catalyst amount, and the MB concentration under visible light irradiation.

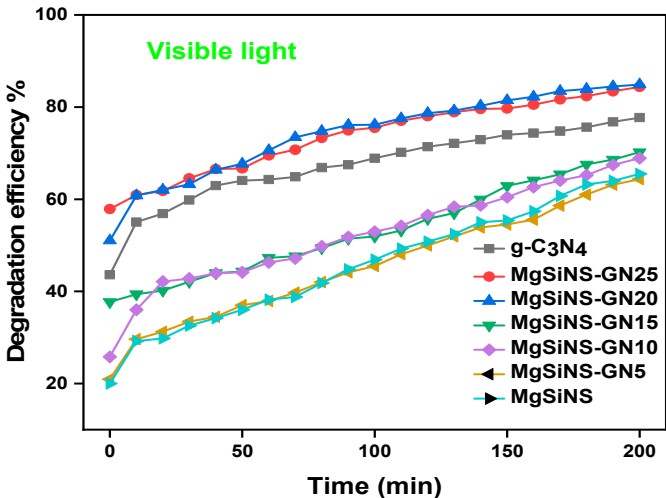

**Figure 7.** The MB photodegradation performance over time using visible light for all the prepared samples.

Some reaction conditions could influence the rate of photodegradation, such as the concentration of dye, catalyst amount, photoenergy exposure time, intensity of light, oxidizing agent, and radical scavengers. To study the effect of pH on the degradation performance of synthesized composites, the pH of the MB solution was adjusted by adding a diluted solution of 0.1 N NaOH or 0.1 N HCl. Different solutions with pH of 4, 6, 8, or 10 were chosen to examine the optimum pH for the MgSiNS-GN20 photocatalyst under visible light. The MB degradation efficiency was calculated at three selected reaction times, i.e., 0, 100, and 200 min, and the obtained results are shown in Figure 8. The MgSiNS-GN20 composite exhibited excellent degradation efficiency (92%) with all examined solutions with different pH levels after 200 min of reaction. It was observed that the MgSiNS-GN20 sample exhibited poor MB physical adsorption (16%) in solution with pH 4, but the photocatalytic degradation efficiency increased fivefold (84%) after 100 min of reaction under visible light. This observation reveals that visible light irradiation plays a critical role in MB degradation in the presence of the MgSiNS-GN photocatalyst. It is interesting to notice that the samples showed high photocatalytic degradation performance with the high basic MB solutions (pH = 8, 10). This is possibly due to the presence of hydroxyl radicals (OH$^*$) assisting the MB photodegradation.

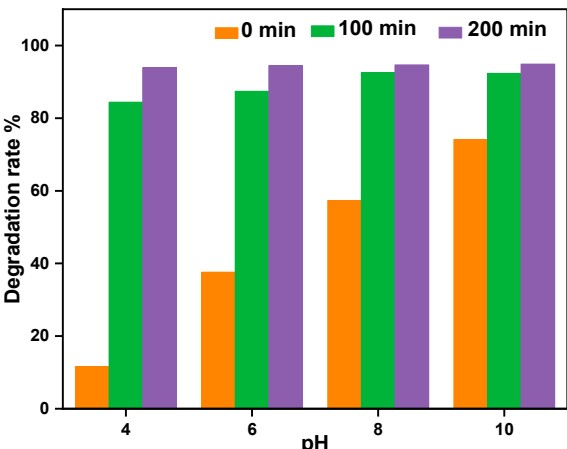

**Figure 8.** The MB degradation efficiency over the MgSiNS-GN20 nanocomposite at different pH solutions using visible light irradiation.

The influence of photocatalyst dosage also was evaluated by using 0.04, 0.06, 0.08, and 0.1 g of MgSiNS-GN300 under visible light. The pH of the concentration of MB was maintained at 10 ppm. The MB photodegradation rate was determined using four different MgSiNS-GN20 amounts at three selected times (0, 100, 200 min) under visible light, and the obtained results are shown in Figure 9a. It is noticeable that an increase in the amount of photocatalyst resulted in an increase in the photodegradation rate, which indicates that the amount of catalyst should be taken into consideration, as it greatly influences the degradation performance. The highest degradation rate was 85% when 0.08 g of catalyst was used. To study the influence of MB concentration on the degradation efficiency of the synthesized nanocomposite (Figure 9b), four different solutions with MB concentrations of 5, 10, 15, and 20 ppm were tested using the MgSiNS-GN20 photocatalyst (0.1 g) at pH 7. The degradation efficiency is high when the MB concentration is 5 ppm (96% after 200 min). The degradation rate decreased to 70% over the MgSiNS-GN20 catalyst with a 20 ppm solution after 200 min. It was observed that the photodegradation efficiency of the catalysts decreased with increasing MB concentration.

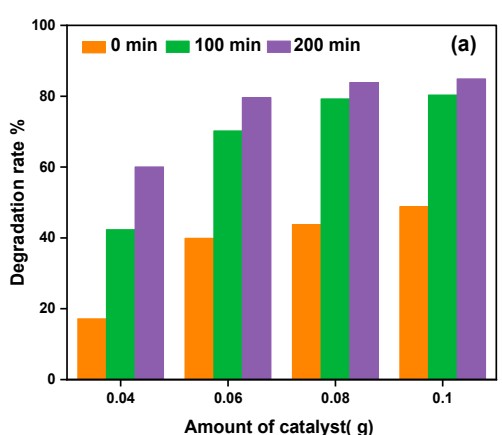

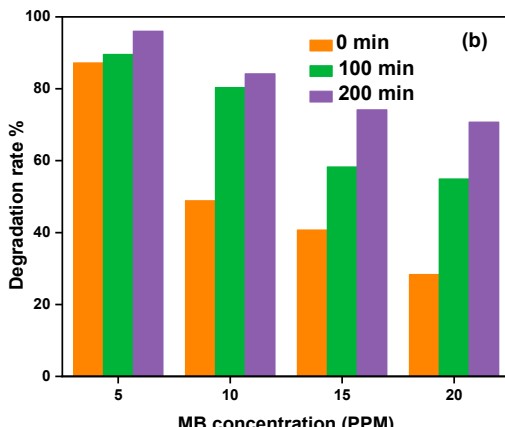

**Figure 9.** (**a**) Influence of amount of MgSiNS-GN20 on the MB photodegradation rate. (**b**) Influence of MB concentration on the MB degradation efficiency under visible light.

Khan et al. [42] published insight on the different photodegradation mechanisms and rate-affecting parameters in the photodegradation of dyes. It is well known that mechanistically, the photodegradation process is based on the production of highly reactive, photo-generated ·OH and ·$O_2^-$ radicals that attack dye molecules and completely degrade them into nontoxic products ($CO_2$ and $H_2O$). A non-catalytic photolysis of MB is also a possible pathway; however, it is impractical to remove MB from aqueous solution using

this method. The possible reaction mechanism of degrading MB over the prepared MgSiNS-GN composites could be explained based on the physicochemical characterization results. Based on the results from band gap energy calculations, a plausible MB photodegradation mechanism is depicted in Scheme 1. The CB and VB potentials for g-$C_3N_4$ and MgSiNS are −1.24 eV and 1.64 eV, and 3.83 eV and 8.19 eV, respectively. The transition of photoelectrons from VB to CB happens in the g-$C_3N_4$ semiconductor under visible light irradiation. The photo-generated holes can transfer from the VB of MgSiNS to g-$C_3N_4$ because the VB potential of g-$C_3N_4$ is more negative compared to the VB potential of MgSiNS, and the CB potential of MgSiNS is more positive compared to the CB level of g-$C_3N_4$. Simultaneously, the photo-generated electrons in the CB of g-$C_3N_4$ could transfer to the CB of MgSiNS, and these electrons could reduce $Mg^{2+}$ species to $Mg^+$ in the MgSiNS. The $Mg^+$ ions on the surface can be re-oxidized into $Mg^{2+}$ through the reaction with oxygen to generate the superoxide radical ($\cdot O_2^-$). The $\cdot O_2^-$ radical can react with $H_2O$ molecules to form $\cdot OH$ radicals [43]. The generated $\cdot OH$ and $\cdot O_2^-$ radicals are active species to degrade MB into $CO_2$, $H_2O$, and $NH_3$ under visible light. Therefore, it can be stated that combining g-$C_3N_4$ with MgSiNS decreases the recombination of photo-induced $h^+$ and $e^-$ through reduction or oxidation reactions on the surface of MgSiNS-GN composites under visible light.

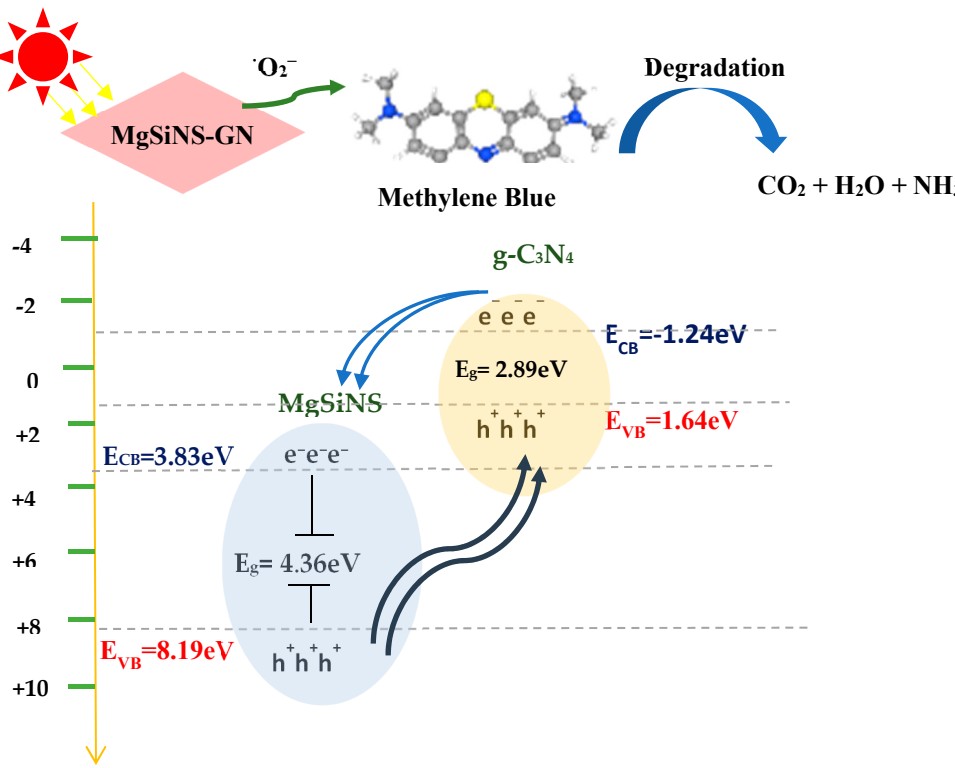

**Scheme 1.** Plausible photodegradation mechanism over synthesized MgSiNS−GN composite.

The reusability of the selected composite, MgSiNS−GN20, was evaluated four times for photodegradation of MB under visible light due to its excellent efficiency. After the first cycle, the MgSiNS-GN20 photocatalyst was separated from the MB solution by centrifugation, and the separated photocatalyst was used for the next cycle. This process was repeated three more times to reach four complete cycles, and the degradation efficiency of the MgSiNS-GN20 catalyst is presented in Figure S4. It is observed that the MB degradation efficiency is gradually decreased with the increase in cycle number, as the in the first cycle, the efficiency is 93%, and it decreased to 85% in the second cycle. It further decreased to 77% and 64% after the third and fourth cycles, respectively. This is probably due to the loss of catalyst amount and deposition of residual MB degradation molecules on the surface of the photocatalyst. The efficiency of the nanocomposite to degrade the MB is still satisfying for reuse, with an estimated degradation efficiency of more than 63%. The

obtained results indicate that the prepared MgSiNS-GN composites can be utilized for the photodegradation of organic pollutants under visible light.

The adsorption capacities of $Pb^{+2}$ ions for all the prepared samples were calculated every 30 min for a total time of 150 min, and the obtained results are shown in Figure 10. The MgSiNS sample showed a remarkable ability to adsorb $Pb^{+2}$ ions compared to other prepared adsorbents; the adsorption capacity of $Pb^{+2}$ ions using MgSiNS was estimated as 0.05 mmol/g after 150 min due to the ion exchange ability of MgSiNS material. In contrast, the $Pb^{+2}$ adsorption capacity for g-$C_3N_4$ did not exceed 0.001 mmol/g, probably due to the weak interaction between g-$C_3N_4$ and $Pb^{2+}$ ions. Interestingly, the MgSiNS-GN composites exhibited lower $Pb^{2+}$ adsorption capacity compared to pure MgSiNS. The increasing order of $Pb^{+2}$ adsorption capacity of composites is MgSiNS-GN25 < MgSiNS-GN20 < MgSiNS-GN15 < MgSiNS-GN10 < MgSiNS-GN5. The MgSiNS-GN20 composite exhibited high efficiency for MB photodegradation, but it showed less $Pb^{+2}$ adsorption compared to the bulk MgSiNS sample. It is considered that the adsorption mechanism of magnesium silicate is influenced by both the electrostatic attraction and the ion exchange for methylene blue. This is mainly due to the presence of multilayers of g-$C_3N_4$ nanosheets on the surface of magnesium silicate, blocking the adsorption sites of magnesium silicates. A decrease in $Pb^{+2}$ adsorption capacity coincided with an increase in g-$C_3N_4$ loading; this is possibly due to the blockage of adsorption sites of MgSiNS by the g-$C_3N_4$. Although the synthesized nanocomposites exhibit less $Pb^{2+}$ adsorption capacity compared to MgSiNS, they possessed adequate $Pb^{2+}$ adsorption capacity with high MB photocatalytic degradation efficiency under ambient conditions. Solehudin et al. [44] synthesized g-$C_3N_4$-urea-melamine isotype heterojunction and observed that the synthesized material exhibits sheet-like morphology with high porosity on the surface of the sample, which increased the photo-generated charge separation and enhanced the photocatalytic activity. Similarly, the MgSiNS-GN composites consisted of nanosheets of both magnesium silicate and g-$C_3N_4$ with mesoporosity. Thus, the morphology and porosity of the MgSiNS-GN composites assisted the enhanced photodegradation and adsorption performance.

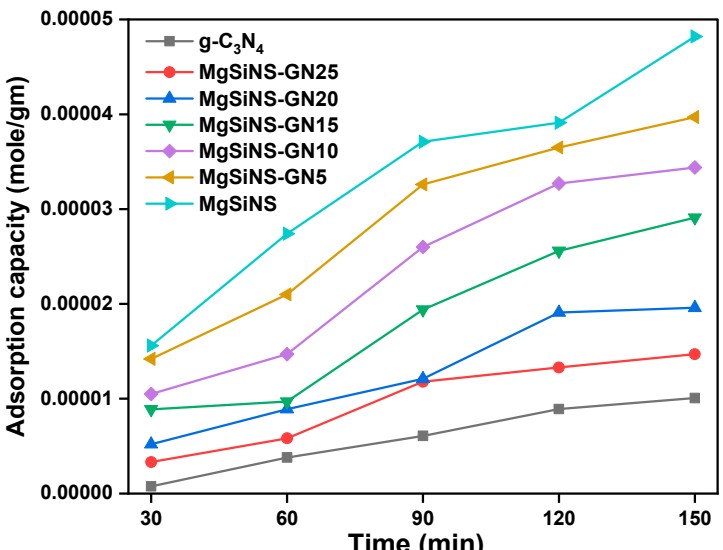

**Figure 10.** $Pb^{+2}$ adsorption capacity of g-$C_3N_4$, MgSiNS, and MgSiNS-GN composites after 150 min.

The influence of the pH of the solution on $Pb^{2+}$ adsorption on the MgSiNS-GN20 composite is shown in Figure 11a. The $Pb^{2+}$ adsorption increased gradually as the pH increased from 2 to 5, and then remained constant. The low adsorption of $Pb^{2+}$ in acidic solution could be attributed to the high concentration of $H^+$ ions, which could compete with $Pb^{2+}$ ions for the active adsorption sites on the surface of MgSiNS-GN20. The pHPZC value of MgSiNS-GN20 is around 5.0, indicating that the adsorbent had a positive surface charge under the test conditions. As the solution pH decreased, the electrostatic repulsion

between the positively charged $Pb^{2+}$ and adsorbents increased, leading to the decline in removal efficiency. At pH 5.0, the $Pb^{2+}$ adsorption of MgSiNS-GN20 is 0.05 m mole/g. It is known that Pb exhibits different species at different pH levels, and when the pH of the solution is greater than 6, the number of soluble Pb ions decreases rapidly. When the pH value is much higher (around 9), the $Pb^{2+}$ ions in the solution disappear. It was reported that $Pb^{2+}$ ions precipitated due to the formation of lead hydroxide at pH higher than 6 [45].

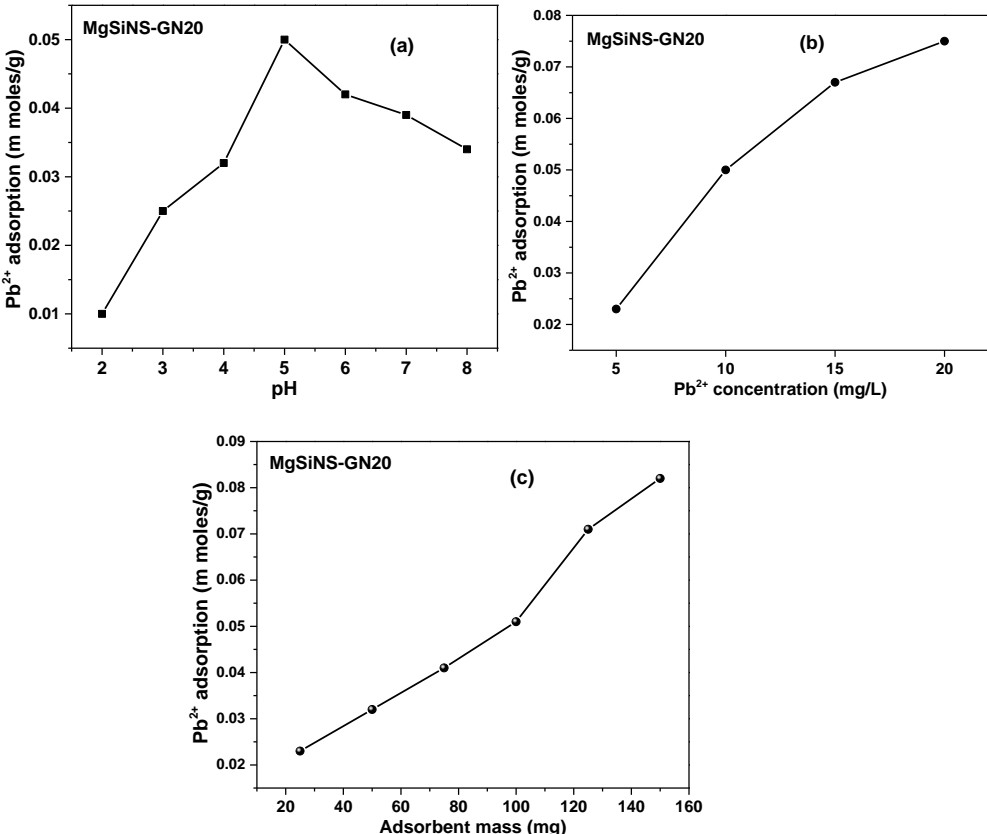

**Figure 11.** (**a**) Influence of pH, (**b**) $Pb^{2+}$ concentration, and (**c**) adsorbent mass on the $Pb^{2+}$ adsorption capacity of the MgSiNS-GN20 composite.

The obtained results reveal that the optimum pH for $Pb^{2+}$ ion adsorption over MgSiNS-GN20 is 5.0 due to competitive adsorption between $H^+$ and $Pb^{2+}$ ions at low pH and the formation of Pb precipitate at high pH. Figure 11b shows the influence of initial $Pb^{2+}$ ion concentration on MgSiNS-GN20′s ability to adsorb $Pb^{2+}$ ions. Four different solutions with different concentrations (5 mg/L, 10 mg/L, 15 mg/L, and 20 mg/L) were used to study the influence of $Pb^{2+}$ ion concentration on the adsorption ability of MgSiNS-GN20 under optimized conditions. The $Pb^{2+}$ adsorption gradually increased with increasing concentration of $Pb^{2+}$ solution. The highest $Pb^{2+}$ removal was observed with a solution of 20 mg/L. The adsorption efficiency decreases with the increase in the concentration of $Pb^{2+}$ ions, as the adsorption sites in MgSiNS-GN20 become saturated. Figure 11c shows the influence of adsorbent mass on the ability of MgSiNS-GN20 adsorbent for $Pb^{2+}$ adsorption. To study this aspect, the mass of the adsorbent was varied from 50 mg to 150 mg under optimized conditions. The $Pb^{2+}$ ion adsorption was increased with the increase in the mass of the adsorbent, as the 0.05 m mole/g of $Pb^{2+}$ was adsorbed when the mass of the adsorbent was 100 mg, and $Pb^{2+}$ ion adsorption increased when 200 mg of MgSiNS-GN20 was used. The observations reveal that $Pb^{2+}$ adsorption is directly proportional to the mass of the adsorbent.

The comparison of MB photodegradation activity of the most active catalyst in this study (MgSiNS-GN20) is compared with some of the previously reported photocatalysts

(Table S1). From Table S1, it is clear that some g-$C_3N_4$-containing photocatalysts such as $La_2O_3$/g-$C_3N_4$, AgCl/$Ag_3PO_4$/g-$C_3N_4$, and $CeO_2$/g-$C_3N_4$ catalysts offered higher MB photodegradation activity (99–100%) compared to MgSiNS-GN20 (93%); however, the synthesized composites showed relatively high photocatalytic activity, along with good $Pb^{2+}$ adsorption ability. This was not feasible with previously reported photocatalysts.

## 3. Experimental

### 3.1. Materials

Magnesium nitrate hexahydrate $Mg(NO_3)_2.6H_2O$, sodium metasilicate nonahydrate [$Na_2SiO_3.9H_2O$], tetraethylammonium hydroxide ($[(C_2H_5)_4NOH]$, TEAOH), urea [$CO(NH_2)_2$], ethanol [$CH_3CH_2OH$], methylene blue [$C_{16}H_{18}ClN_3S$], and lead nitrate [$Pb(NO_3)_2$], all analytical-grade reagents, were purchased from Sigma-Aldrich (Saint Louis, MO, USA) and utilized as received.

### 3.2. Synthesis of Nanomaterials

#### 3.2.1. Magnesium Silicate Nanosheets (MgSiNS)

To synthesize magnesium silicate nanosheets (MgSiNS), a simple solvothermal synthesis method was adapted. Initially, 15.4 g of $Mg(NO_3)_2.6H_2O$ was placed in a 500 mL beaker and dissolved in 100 mL of mixed solvent (distilled water and ethanol, 1:1 vol ratio). Another solution was prepared by dissolving 34.5 g of $Na_2SiO_3.9H_2O$ in 200 mL of mixed solvent. The sodium silicate solution was added to magnesium nitrate solution under magnetic stirring for 15 min. Then, 15 mL of TEAOH solution was added dropwise to obtain a white slurry. After complete addition, the contents were left under magnetic stirring at room temperature for 6 h to ensure complete reaction between the solutions. Then, the resulting solution was transferred into a 500 mL Teflon-lined autoclave, and the tight autoclave was placed in an electrical oven at 90 °C for 24 h. After completion of solvothermal treatment, the obtained, white-colored product was washed with water and ethanol several times. Then, centrifugation was performed to separate the product after washing and the product was dried in an electrical oven at 80 °C for three hours to remove water and ethanol. Finally, the product was calcined at 500 °C in a muffle furnace for four hours.

#### 3.2.2. Graphitic Carbon Nitride (g-$C_3N_4$)

The graphitic carbon nitride (g-$C_3N_4$) nanosheets were synthesized by following a modified method previously reported [22]. A calculated amount of urea was placed in a fitted ceramic crucible and heated gradually to 400 °C for two hours without additional flowing air in a muffle furnace. After thermal treatment, the resulting yellowish g-$C_3N_4$ powder was cooled down at room temperature and used without any further treatment.

#### 3.2.3. MgSiNS/g-$C_3N_4$ Nanocomposites

The MgSiNS/g-$C_3N_4$ nanocomposites with different compositions were synthesized. Calculated amounts of MgSiNS and g-$C_3N_4$ were dispersed in 60 mL of mixed solvent (water and ethanol) and subjected to ultrasonication treatment for one hour at 25 °C. After sonication treatment, the nanocomposites were centrifugated and dried at 90 °C for 24 h to evaporate the excess solvent. Then, the samples were calcined at 300 °C to ensure complete evaporation of water and ethanol for further use. Table 2 shows the elemental composition of nanocomposites determined by ICP-AES analysis. A minor difference between nominal and actual loadings of MgSiNS and g-$C_3N_4$ was observed.

### 3.3. Characterization of Samples

To examine the morphological and chemical properties of synthesized materials, various analytical and spectroscopic instruments were utilized. The XRD patterns were obtained using a Bruker D8 Advance device with a monochromator, Cu Kα radiation ($\lambda$ = 1.5405Å) at 40 kV and 40 mA. To evaluate FT-IR spectra in the wavenumber range

of 400–4000 cm$^{-1}$, a Perkin-Elmer FT-IR spectrometer was used. The TEM images of the synthesized samples were obtained using a JEOL JEM-2100 transmission electron microscope. The XPS analysis of the samples was performed using a SPECS GMBH X-ray photoelectron spectrometer using Mg Kα (1253.6 eV) radiation at the pressure of $5 \times 10^{-9}$ mbar. The N$_2$ physical adsorption experiments were accomplished by using a Quantachrome Autosorb-1 apparatus. Outgassing of the samples was carried out at 200 °C to ensure the removal of moisture and other unwanted adsorbed species on the surface of the samples. The N$_2$ isotherm data were used to obtain the surface area and porosity of the prepared samples. Diffuse reflectance UV–vis spectra of the samples were collected using a Thermo-Scientific evolution UV–vis spectrophotometer in the wavelength range of 100–800 nm.

**Table 2.** Composition of MgSiNS and g-C$_3$N$_4$ nanocomposites.

| Composite | MgSiNS (wt %) | g-C$_3$N$_4$ (wt %) | * Elemental Composition (Mass %) | | | | |
|---|---|---|---|---|---|---|---|
| | | | Mg | Si | O | C | N |
| MgSiNS-GN5 | 5 | 95 | 20.8 | 22.2 | 44.4 | 5.9 | 6.7 |
| MgSiNS-GN10 | 10 | 90 | 18.3 | 19.5 | 40.5 | 10.5 | 11.2 |
| MgSiNS-GN15 | 15 | 85 | 16.5 | 17.7 | 35.3 | 14.8 | 15.7 |
| MgSiNS-GN20 | 20 | 80 | 14.6 | 16.1 | 32.1 | 18.1 | 19.1 |
| MgSiNS-GN25 | 25 | 75 | 12.7 | 13.2 | 30.4 | 20.4 | 22.3 |

* ICP-AES and CHN analyses.

### 3.4. Photocatalytic Degradation of Methylene Blue

In brief, the photocatalytic degradation experiments were performed in a laboratory-built reactor. In the typical procedure, 100 mL of 10 ppm MB solution was placed in a round bottom flask, then 0.1 g of the prepared photocatalyst was added to the solution and stirred using a magnetic stirrer in the dark for 30 min before turning on the visible light to start the photodegradation. The round bottom flask was then evacuated and irradiated by a 300 W Xe lamp providing a flux of approximately 125 mW cm$^{-2}$ in the reaction zone. During the reaction, the liquid samples were withdrawn using a filter syringe (0.45 μm) every 10 min to avoid suspended catalyst particles. The change in the MB concentration in collected samples was analyzed using a UV–vis spectrometer (Thermo Fisher Scientific Evolution 160 nm) in the wavelength range of 400 to 800 nm. The degradation percentage was calculated according to the following formula [46]:

$$\text{Degradation\%} = \left(1 - \frac{A}{A_0}\right) \times 100 \qquad (1)$$

where $A_0$ is the adsorption of methylene blue (MB) before the degradation and $A$ is the adsorption of MB in a certain time.

### 3.5. Pb$^{2+}$ Adsorption Studies

In the typical experiment, 100 mg of nanocomposite sample was added to 100 mL of 10 mg/L of Pb$^{+2}$ solution under constant stirring, and the pH of the solution was maintained at 6.5. The liquid samples were withdrawn periodically every 30 min using a syringe with a filter (0.45 μm) to avoid any suspended adsorbent particles in the solution. An inductively coupled plasma mass spectrometer (ICPMS) was used to determine the concentration of Pb$^{2+}$ solutions. The adsorption capacity of adsorbents was calculated using the following expression: $q = (C_o - C/m) \times V$, where $q$ (mol/g) is the adsorption capacity, $C_o$ is the initial concentration of Pb$^{+2}$ in ppm, $C$ is the concentration (mg/L) in a certain time, and $V$ and $m$ are the volume of Pb$^{+2}$ solution in (L) and the weight of adsorbent in (g), respectively.

## 4. Conclusions

Pure MgSi and g-C₃N₄ nanosheets and their composites (MgSiNS-GN) with different weight ratios were synthesized in this study. The synthesized materials were characterized with various characterization techniques. The XRD, FT-IR, TEM, and XPS characterization results revealed that the structure, morphology, and electronic properties of nanocomposites are in agreement with features of bulk MgSiNS and g-C₃N₄ materials. The prepared composites showed superior optical properties and low band gap energy compared to parent materials, demonstrating the usefulness of composites for photocatalysis. Optimization of g-C₃N₄ weight percent was achieved, as the 20 wt % sample exhibited the lowest band gap energy among all the synthesized samples. The photocatalytic MB degradation activities suggested that MgSiNS-GN20 presented superior performance to degrade MB under visible light irradiation compared to other composites including MgSiNS and g-C₃N₄ due to its optimal band gap and physicochemical properties. The change in the pH of the MB solution played a crucial role in the degradation of MB, as an increase in the pH value enhanced the photodegradation rate. Increasing the catalyst amount led to faster degradation; however, the increase in organic pollutant concentration lowered the degradation efficiency. In addition, the synthesized MgSiNS-GN nanocomposites possessed suitable $Pb^{2+}$ adsorption ability. The MgSiNS-GN20 composite offered high $Pb^{2+}$ adsorption of 0.005 mol/g and excellent MB degradation efficiency of 93% at pH 7 within 200 min compared to other composites. Inexpensive, eco-friendly, and recyclable, magnesium silicate and graphitic carbon nitride composites can be promising materials to remove organic and inorganic contaminants from water.

**Supplementary Materials:** The following supporting information can be downloaded at: https://www.mdpi.com/article/10.3390/catal12101256/s1, Figure S1: Tauc plots for all the synthesized materials, Figure S2: Deconvoluted XPS spectra for MgSiNS (a) Mg*1s*, (b) Si*2p* and (c) O*1s*, Figure S3: Deconvoluted XPS spectra for g-C₃N₄ (a) C*1s* (b) O*1s* and (c) N*1s*, Figure S4: Recycling of MgSiNS-GN20 for photodegradation of MB, Table S1: Comparison of photocatalytic MB degradation performances of different catalysts [47–56].

**Author Contributions:** Data curation, M.A.A. and K.N.; formal analysis, A.A.; investigation, A.A. and M.A.A.; methodology, M.A.A. and A.A.; project administration, A.A. and K.N.; resources, K.N.; supervision, K.N.; writing–original draft, M.A.A.; writing–review & editing, K.N. All authors have read and agreed to the published version of the manuscript.

**Funding:** This research received no external funding.

**Data Availability Statement:** The data presented in this study is available in Supplementary Material.

**Acknowledgments:** Muhmmed Ali Alnassar acknowledge financial support from Qassim University for his M.Sc. studies.

**Conflicts of Interest:** The authors claim that they do not have any known conflicting financial interest or personal affiliations that may seem to have impacted the work presented in this study.

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
