# Peer review of "Visible Light Active Magnesium Silicate–Graphitic Carbon Nitride Nanocomposites for Methylene Blue Degradation and Pb2+ Adsorption"

_catalysts, doi:10.3390/catal12101256_

Round 1

Reviewer 1 Report

Comments to the authors

The manuscript reports synthesis of Magnesium silicate nanosheets and graphitic carbon nitride (g-C3N4) nanocomposites and teir utilization for dye degradation and metal adsorption. Production of an effective and economical materials for pollutants remediation have received increased attention in the last few decades. The present work affords a promising strategy for the designing effective and economical composite material for dye degradation as well as metal ion adsorption. I suggest major revision for quality enhancement, which are given bellow.

11.      The manuscript contains huge grammatical and typographical mistakes which need correction and the language need sufficient improvement.

22.      Line 37, what does “metritis” means here?

33.      Discuss some toxic effects of methylene blue in the introduction section and photodegradation parameters from this review and cite this: Water 2022, 14, 242. https://doi.org/10.3390/w14020242.

44.      Keep all the figures after their discussion.

55.      The authors should properly correlate the XRD and FTIR data with the results presented in the manuscript.

66.      Mention and discuss Figure 5 as figure 5a and 5b.

77.      How the authors distinguished between adsorption and degradation of dyes through catalysts.

88.      The radicals should be mentioned through dot (.) instead of *.

99.      Explain the mechanism of parameters of photodegradation using and citing this review article Journal of Environmental Chemical Engineering 8 (2020) 104364.

110.  How was the catalysts recovered?

111.  Mention some toxic effect of lead (Pb2+) in the introduction section.

112.  The authors mentioned in the abstract that the MgSiNS-GN20 composite exhibited high degradation and adsorption efficiency while in the adsorption this nanocomposite displayed les adsorption. Clarify this.

113.  In conclusion discuss the obtained scientific results.

114.  All the cited references are old. Add some updated references.

Author Response

Detailed responses to reviewer comments are in the attached file

Reviewer 2 Report

Reviewer’s comments

Manuscript Number: catalysts-1921578

Title: Visible light active magnesium silicate-graphitic carbon nitride nanocomposites for methylene blue degradation and Pb2+ adsorption

Journal: Catalysts

The work reports methylene blue degradation and Pb2+ adsorption using MgSiNS and g-C3N4 nanocomposites. The manuscript is well organized, and the material is fully characterized. However, some revisions are needed as follows:

  1. Since the prepared materials are nanocomposites, the elemental composition (Mg:Si:C:N:S) should be provided using EDX of ICP data to make sure the experimental ratios are agree with the nominal ratios.
  2. MB removal was done via photocatalytic degradation, while Pb2+ was via adsorption. These are two different mechanisms, thus focusing on only one study is recommended.
  3. The findings should be compared with other related materials. A detailed table of comparison should be provided.
  4. The aim of this study is not clear. The novelty should be clearly mentioned.
  5. The provided reference [27] is not the correct source of Scherrer equation, citing the original source is required.
  6. The correlation between structural/morphological findings and catalytic performance should be discussed. Here some literature could help: Advanced Powder Technology 2020; 31: 1891-1902, Materials Chemistry and Physics 2020; 242: 122520, Journal of Molecular Structure 2022; 1250: 131800.
  7. For Pb2+, other experimental parameters are needed to be investigated, such as dose, pH, concentration, etc.

Author Response

(The authors gave the same response as above.)

Round 2

Reviewer 1 Report

Sufficients modifications were carried out and thus the paper can be accepted after minor spell an grammer checking. Also put all the fiures after their text discussion.

Reviewer 2 Report

Some of the comments are addressed, howere, still some of them need to be considered as well:

1.    The provided reference [27] is not the correct source of Scherrer equation, citing the original source is required.

2.    For Pb2+, other experimental parameters are needed to be investigated, such as dose, pH, concentration, etc.
